# Machine learning predicts translation initiation sites in neurologic diseases with nucleotide repeat expansions

**Alec C. Gleason[1], Ghanashyam Ghadge[1,2], Jin Chen[3], Yoshifumi Sonobe[1,2], Raymond P. Roos[1,2] ***

**1** University of Chicago, Chicago, Illinois, United States of America, **2** Department of Neurology, University of Chicago, Chicago, Illinois, United States of America, **3** Department of Pharmacology, University of Texas Southwestern Medical Center, Dallas, Texas, United States of America

* rroos@neurology.bsd.uchicago.edu

## Abstract

A number of neurologic diseases associated with expanded nucleotide repeats, including an inherited form of amyotrophic lateral sclerosis, have an unconventional form of translation called repeat-associated non-AUG (RAN) translation. It has been speculated that the repeat regions in the RNA fold into secondary structures in a length-dependent manner, promoting RAN translation. Repeat protein products are translated, accumulate, and may contribute to disease pathogenesis. Nucleotides that flank the repeat region, especially ones closest to the initiation site, are believed to enhance translation initiation. A machine learning model has been published to help identify ATG and near-cognate translation initiation sites; however, this model has diminished predictive power due to its extensive feature selection and limited training data. Here, we overcome this limitation and increase prediction accuracy by the following: a) capture the effect of nucleotides most critical for translation initiation via feature reduction, b) implement an alternative machine learning algorithm better suited for limited data, c) build comprehensive and balanced training data (via sampling without replacement) that includes previously unavailable sequences, and d) split ATG and near-cognate translation initiation codon data to train two separate models. We also design a supplementary scoring system to provide an additional prognostic assessment of model predictions. The resultant models have high performance, with ~85–88% accuracy, exceeding that of the previously published model by >18%. The models presented here are used to identify translation initiation sites in genes associated with a number of neurologic repeat expansion disorders. The results confirm a number of sites of translation initiation upstream of the expanded repeats that have been found experimentally, and predict sites that are not yet established.

**Data Availability Statement:** Source code, data, as well as models are accessible at https://github.com/Agleason1/TIS-Predictor. A DOI was assigned

to the repository using Zenodo: http://doi.org/10.5281/zenodo.5110255.

**Funding:** The author(s) received no specific funding for this work.

**Competing interests:** The authors have declared that no competing interests exist.

**Abbreviations:** RAN, Repeat-associated non-AUG; RLI, Repeat length-independent; KCS, Kozak consensus sequence; KSS, Kozak similarity score; AUROC, Area under receiver operating characteristic; ROC, Receiver operating characteristic; RFC, Random forest classifier.

# Introduction

## Background

More than 40 neurologic diseases are caused by expansions of repeat nucleotide sequences in causative genes. The repeats range from three nucleotides, such as 'CTG' associated with myotonic dystrophy Types I and II, to up to 12 nucleotides, such as 'CCCCGCCCCGCG', associated with progressive myoclonus epilepsy. Protein products translated from expanded repeat sequences tend to accumulate and aggregate, and have been proposed to contribute to disease [1–10]. Interestingly, in some cases, the repeats have been shown to be translated in all three reading frames from both the plus and minus strands of the RNA [11] by a process termed repeat-associated non-AUG (RAN) translation.

It is believed that the affinity of translational machinery to folded regions of the RNA may underlie translation of the repeat sequences, and that this may occur from sequences in a repeat length-independent (RLI) mechanism. Sequences may be ordered in such a way that they naturally increase the affinity of translational machinery to initiate at a particular codon. In such a process, translation may initiate not only within the repeat region, but also from sites upstream of the repeat sequences. In this case, repeat peptides will be produced if a stop codon is not encountered by the translational machinery before encountering the repeats. The large number of nucleotides that comprise and precede repeat sequences make the identification of RLI translation initiation sites challenging without application of experimental results or computational methods.

## Novelties and contributions of the proposed work

Since finding translation initiations sites experimentally may be difficult, it is valuable to narrow testing to specific codons likely to initiate translation. Although predictive models could potentially identify such codons, machine learning has never been used to help locate these sites in neurologic disorders. One likely reason is a lack of sufficient experimental data (especially human data) with confirmed near-cognate translation initiation sites for model training. While several models for predicting ATG initiation sites have been proposed, they are not applicable for neurologic disorders as they do not predict near-cognate codons. In fact, there may only be one model (called TITER) trained exclusively on human data for predicting whether an ATG or near-cognate codon initiates translation [12]. TITER addresses limitations of an earlier model that predicts whether a codon initiates translation [12, 13]. Unfortunately, the large feature selection of TITER and limited training data may impair its predictive accuracy. In addition, there are no provided tools for TITER to predict all translation initiation sites in a given sequence at once or to evaluate the strength of each prediction.

Here we describe predictive models that have ~85–88% accuracy, exceeding that of TITER by >18%. Our models reduce feature selection to capture the effect of ten critical nucleotides that flank both sides of a putative translation initiation codon since they have an important impact on translation initiation [14–21]. One model is tailored for ATG and the second for near-cognate codons because of their differences in initiating translation [22, 23]. These models use an alternative machine learning algorithm better suited for limited data [24]. We implement unbiased training data through sampling techniques *without* replacement using human gene sequences. Furthermore, we create a unique scoring system that allows us to independently evaluate the likelihood of each prediction, in addition to providing the prediction probabilities associated with the models. Our models provide visualization of *all* predicted translation initiation sites in a given sequence with supplemental scores. The models confirm nearly all experimentally established translation initiation sites in genes with nucleotide repeat

expansions that cause neurologic disease, and predict multiple sites that have not yet been investigated.

## Results

### Kozak similarity score algorithm

Before applying machine learning, we evaluated the performance of a more straightforward algorithm that uses a number of nucleotides as predictors of translation initiation. This algorithm was designed to predict the ability of a codon to initiate translation based on the similarity of its surrounding sequence profile to the Kozak consensus sequence (KCS). The KCS is a nucleotide motif that most frequently borders the canonical translation initiation codon (ATG) and optimizes translation initiation at the site. Although there exist slight variations, this motif is typically accepted as a pattern of underlined nucleotides bordering the AUG codon: CCRCCAUGG, with the nucleotide designated by R being a purine, most typically adenine [14]. The sequence logo of the KCS (Fig 1) identifies conserved nucleotides that tend to border ATG codons that initiate translation. The vertical length of each letter in the sequence logo is related to the observed probability for a particular nucleotide to be at a certain position, as well as the impact of the position on the efficiency of translation initiation. It is formulated by the Shannon method [25].

We designed a weighted scoring algorithm based on the KCS sequence logo (Fig 1) and the ten bases preceding and following the codon. Each nucleotide of the 23-base sequence has a value assigned equal to the height of the nucleotide at its respective position, as illustrated in Fig 1. If a nucleotide is not present in a position, it is assigned a value of zero. These values are then summated, and the total divided by the maximal possible summated score had each nucleotide in the sequence been assigned the largest possible value for its position. This division serves to make final values more feasible for interpretation. As opposed to the pre-normalized score range of about 0 to 0.5990, scores derived from the normalization procedure more conveniently range from 0 to 1. Overall, the final output score of a codon, called the

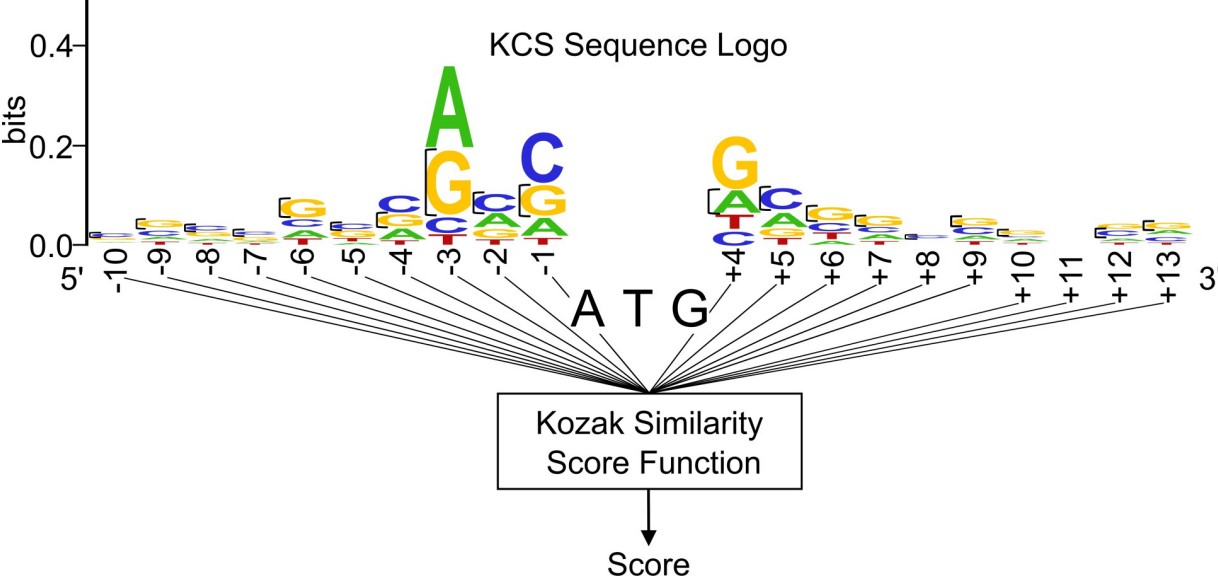

**Fig 1. Schematic of the Kozak similarity score algorithm.** Based on the sequences flanking an input codon, the algorithm references the KCS Sequence Logo to assign the codon a score.

Kozak similarity score (KSS), is deduced by the equation:

$$KSS(codon) = \frac{1}{KSS\_bits_{max}} \sum_{p=1}^{20} bits(nucleotide_p) \qquad (1)$$

In this equation, *p* denotes the position of a nucleotide bordering the codon. Values *p = 1, 2, 3, . . ., 10* designate the positions of the ten nucleotides (from left to right) on the left side of the codon, whereas values *p = 11, 12, 13, . . ., 20* designate the positions of ten nucleotides (from left to right) on the right side of the codon. Furthermore, *bits(nucleotide)* is the assigned height of a particular nucleotide with reference to the KCS sequence logo (Fig 1). $KSS\_bits_{max}$ is the maximum possible value of $\sum_{p=1}^{20} bits(nucleotide_p)$.

We used this algorithm with sequences flanking known instances of ATG translation initiation, and produced a histogram distribution of the resulting scores (Fig 2). We created two baselines in order to compare the scoring of ATG translation initiation codons against ATG codons that do not initiate translation. For the first baseline, we ran the algorithm on one hundred thousand 'dummy' ATG codons that had completely randomized sequences without missing nucleotides (i.e., a randomized adenine, cytosine, thymine, or guanine in every position flanking the codons), and then graphed the resulting score distribution. For the second

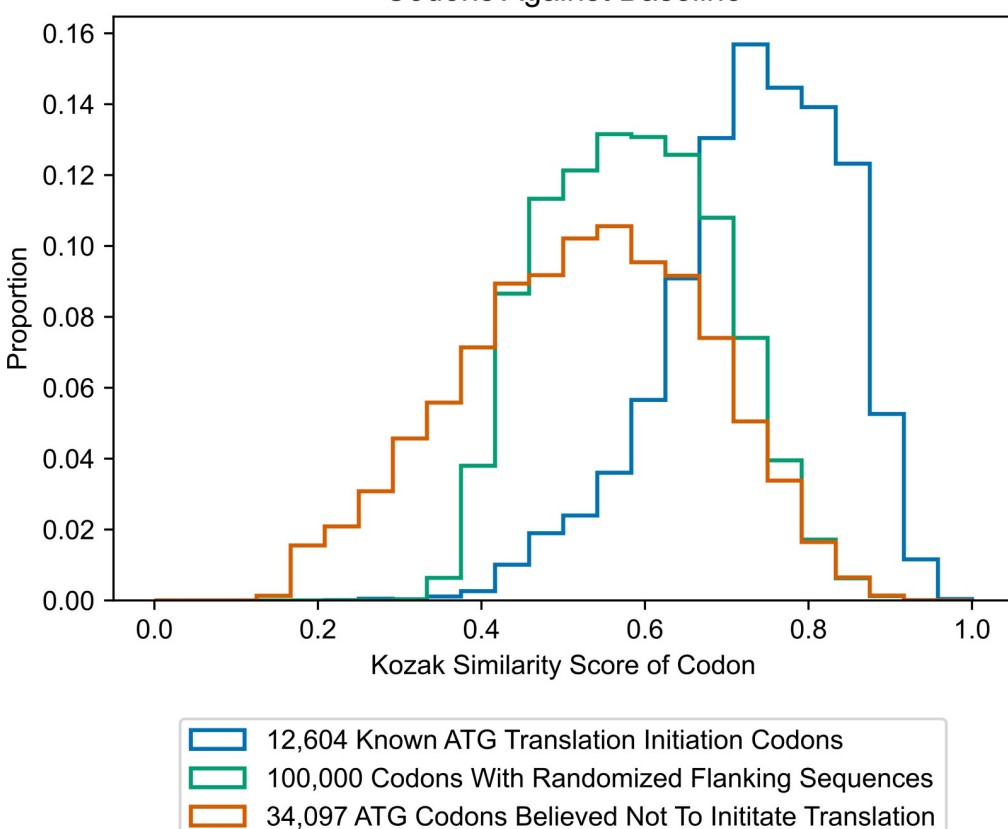

**Fig 2. Kozak similarity scores of ATG translation initiation codons against baseline.**

baseline, we ran the algorithm on a series of ATG codons derived from the human genome that are believed not to initiate translation.

These histograms were generated from a large dataset and therefore they may more accurately serve as representations of algorithm scoring for codon classifications: codons that initiate translation, a mixture of codons that initiate translation and do not initiate translation, and codons that are not believed to initiate translation. Of note, the histogram in Fig 2, which represents a randomized combination of codons that initiate and do not initiate translation, is centered at ~0.59 for both the mean and median. In contrast, the histogram representing ATG codons that initiate translation has a left-skewed distribution, with mean and median scores of about 0.73 and 0.74, respectively. The histogram representing ATG codons expected not to initiate translation has a slightly right-skewed distribution, with mean and median scores of about 0.52 and 0.53, respectively.

Although the exact sequences bordering near-cognate initiation codons have not been identified, as in the case for canonical ATG initiation codons, similarities between the two sequences have been described. For instance, a bioinformatics study that analyzed sequences bordering forty-five mammalian near-cognate initiation codons (including CUG, GUG, UUG, AUA, and ACG codons) found that a guanine or cytosine usually occupies the -6 position (i.e., 6 bases upstream of the codon) [26]. As shown in Fig 1, a guanine or cytosine is also most prevalent in the KCS at this position. The same study also noted the presence of a purine (adenine or guanine) in the -3 position from the codon, which are the two most likely nucleotides to occur in the same position of the KCS [26]. CUG near-cognate codons that most frequently initiate translation usually have an adenine in the -3 position [27]. Although the frequencies of adenine and guanine in the -3 position of the KCS are similar, analysis suggests that adenine is more conserved. For example, if the nucleotide weightings in the KCS are analyzed, adenine is conserved in about 47% of cases at that position versus guanine, with about 37% conservation. Both the bioinformatics study as well as a publication analyzing peptide translation from CUG-initiating mRNA constructs show enhanced translation when guanine is at the +4 position (1 base downstream of the initiation codon) [19, 26]. In the KCS, guanine is also most conserved at the +4 position.

Because of the above similarities, we applied the algorithm to score known near-cognate codons that have been shown to initiate translation (Fig 3). Interestingly, the distributions of all results are left-skewed, visibly differing from results derived from scoring of 'dummy' codons with randomized flanking sequences as well as codons expected not to initiate translation. The distribution of scores for known CTG codons has a mean and median of about 0.69, while the distribution of scores for known GTG codons has a mean and median of about 0.69 and 0.70, respectively. The distribution of scores for known TTG codons has a mean and median of about 0.65. These results are an indication that the KSS of near-cognate codons can be used to predict their ability to initiate translation.

To use the KSS as a predictor of translation initiation ability, a threshold score has to first be determined. In this way, an algorithm could classify codons with a score above the threshold as initiating translation, and below it, not initiating translation. To find the best threshold, virtual simulations were run using different score cutoffs to classify already known ATG initiation codons and ATG codons expected not to initiate translation. Since there are at least 12,603 cases of known ATG initiation codons in contrast to at least 34,097 ATG codons believed not to initiate translation, the data were first balanced. In this way, the cutoff derived would not bias classifications of codons in favor of not initiating translation. Next, all possible cutoff values were set, ranging from 0.580 to 0.700 by increments of 0.001. This range was determined by contrasting distributions in Fig 2. For each of these cutoff values, one thousand simulations were run classifying the data of 12,603 known ATG translation initiation codons

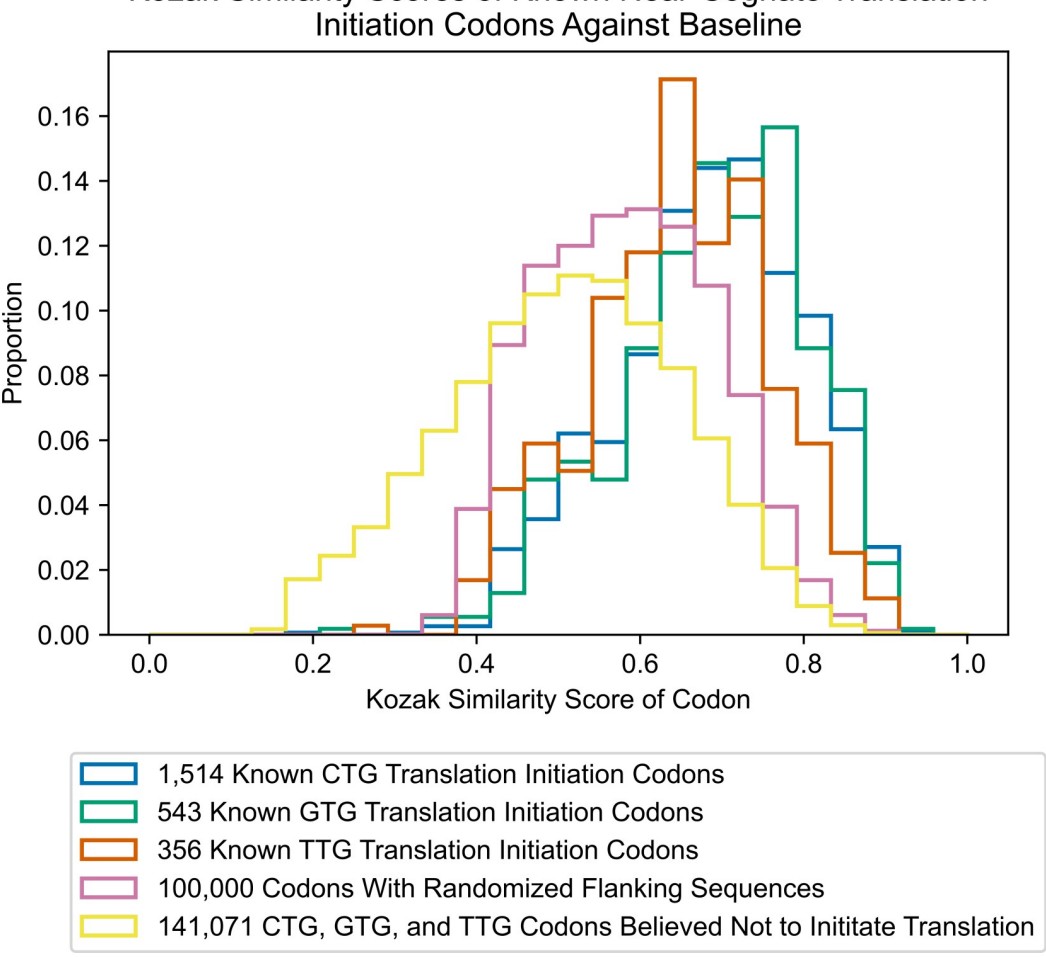

**Fig 3. Kozak similarity scores of near-cognate translation initiation codons against baseline.**

on a randomized subset containing 12,603 of the total 34,097 non-initiating ATGs. Errors were averaged for the one thousand runs at each cutoff value. A cutoff of about 0.64 had the most minimized error. When tested on data containing the 12,603 known ATG-initiating codons and randomized 12,603 instances of non-initiating ATGs, the average accuracy of the model was ~80%.

The area under receiver operating characteristic (AUROC) score from one of the thousand model simulations selected at random was calculated to be 0.876. This score is a useful metric as it indicates the model's discriminatory ability. In the model context, it would correctly assign a greater prediction value for a codon to initiate translation if it indeed were a translation initiation codon 87.6% of the time [28]. A random classifier has a score of 0.5, whereas a perfect classifier has a score of 1.0 [29]. This score is calculated as the area under the ROC curve. This is a graphical illustration of the model's ability to correctly categorize positives (i.e., the true positive rate) against decreased discrimination (increased false positive rates).

As carried out in the case of ATG, the cumulative data of the CTG, GTG, and TTG codons was used to deduce a cutoff value for the algorithm's scoring of all near-cognate codons. To identify the best cutoff for near-cognate codons, the same simulation process was used as was carried out for ATG codons. Using this simulation method, with balanced near-cognate codon

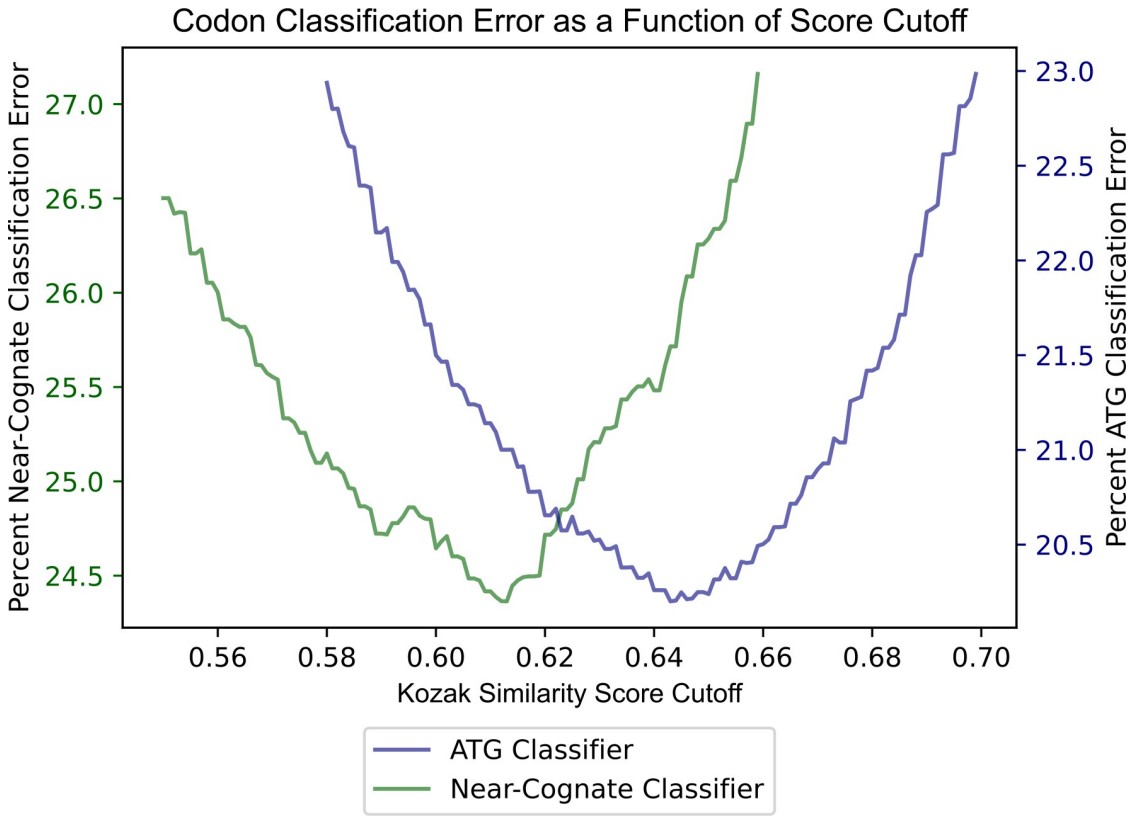

**Fig 4. Error classifying ATG and near-cognate codon ability to initiate translation using Kozak similarity score.**

data consisting of equal numbers of positives (near-cognate initiation codons) and negatives (near-cognate codons that do not initiate translation), the best cutoff of the algorithm classification was about 0.61 for near-cognate codons. After a thousand simulations, the algorithm revealed an average accuracy of about 75.60% for classifying near-cognate codons as initiating translation or not initiating translation. The AUROC score calculated from one randomly selected simulation was 0.835. Classifier accuracy for ATG and near-cognate codons is depicted in Fig 4 as a function of KSS score cutoff. ROC curves for the ATG and near-cognate classifiers are displayed in Fig 5.

## KSS as a reference for likelihood of translation initiation

In the previous section, the weighted scoring algorithm based on the KCS was used as a model to classify whether codons were likely to initiate translation. However, one could question whether the scores of the weighting system could also be used as a metric. To investigate this issue, 12,603 instances of ATGs that initiate translation and 34,097 ATGs believed not to initiate translation were compiled. One thousand balanced test datasets, containing the 12,603 positive ATG instances along with randomly sampled negative ATG instances of the same number, were gathered. The average proportion of codons that initiate translation with a KSS exceeding particular values, across all test datasets was determined. These KSS thresholds ranged from zero to one by increments of 0.02. The proportion of ATGs that initiate translation had a positive correlation with the KSS. In other words, a greater proportion of ATGs would initiate translation with an increased score. This score appeared useful since one could

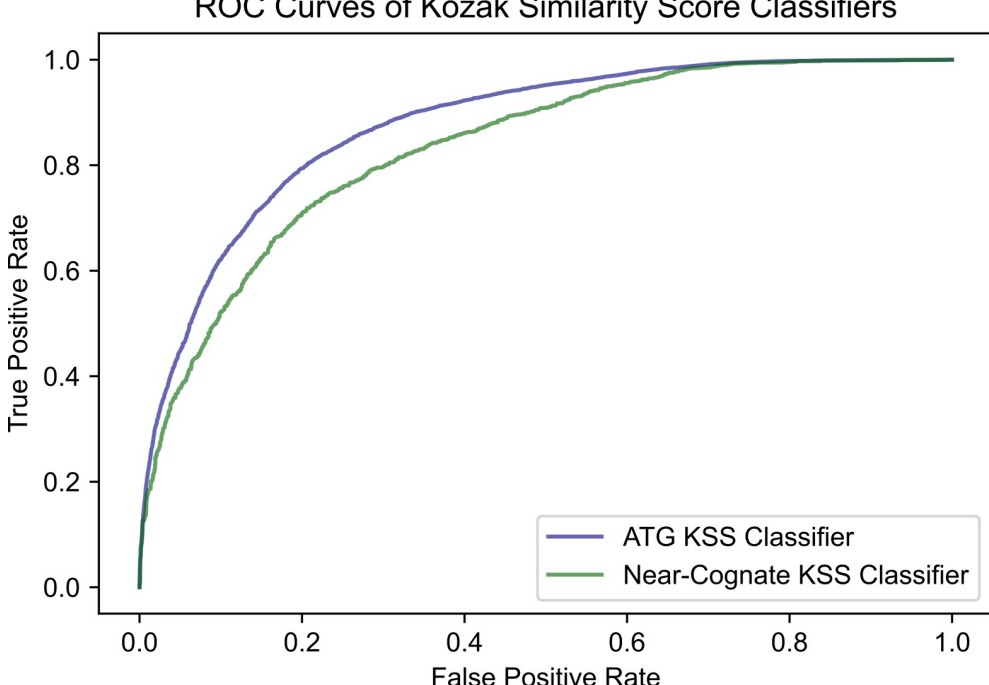

**Fig 5. ROC curves of the ATG and near-cognate Kozak similarity score classifiers.** The AUROC score (area under the curve) of the ATG classifier is 0.876. The AUROC score of the near-cognate RFC is 0.835.

approximate the proportion of ATG codons that initiate translation with a similar KSS related to a particular codon encountered.

The same evaluation was conducted for near-cognate codons to deduce if there was a similar trend. The procedures previously applied to the ATG data were used for the cumulative total of 2,413 instances of near-cognate codons that initiate translation, and 141,071 instances of near-cognate codons believed not to initiate translation. There was a positive correlation between the proportion of near-cognate codons that initiate translation and the KSS. In fact, the trend was quite similar to that obtained for ATG data. The results suggest that the KSS is not limited as a metric for ATG codons, but could be used to estimate the likelihood of a near-cognate codon to initiate translation as well. The results of the analysis for ATG and near-cognate codons is shown in the graph and table of Fig 6.

## Random forest classifiers

A strong and practical approach for identifying translation initiation codons also includes the application of a machine learning model. Machine learning models are powerful, as they can analyze large amounts of complex data, determine patterns and codependences that are difficult to process by a human, and learn from mistakes to improve over time [24]. Although biological pathways are often sophisticated and produce remarkably diverse data, machine learning models can provide direction for such processes that are not completely understood.

We decided to implement a random forest classifier (RFC). This machine learning algorithm typically produces satisfactory results with partly missing data, bears little impact from outliers, and mitigates overfitting. Furthermore, the RFC is a highly preferred model in contemporary genomics [30]. The RFC is based on many decision trees, typically generated from large subsets of data. As each decision tree may split data differently in the classification

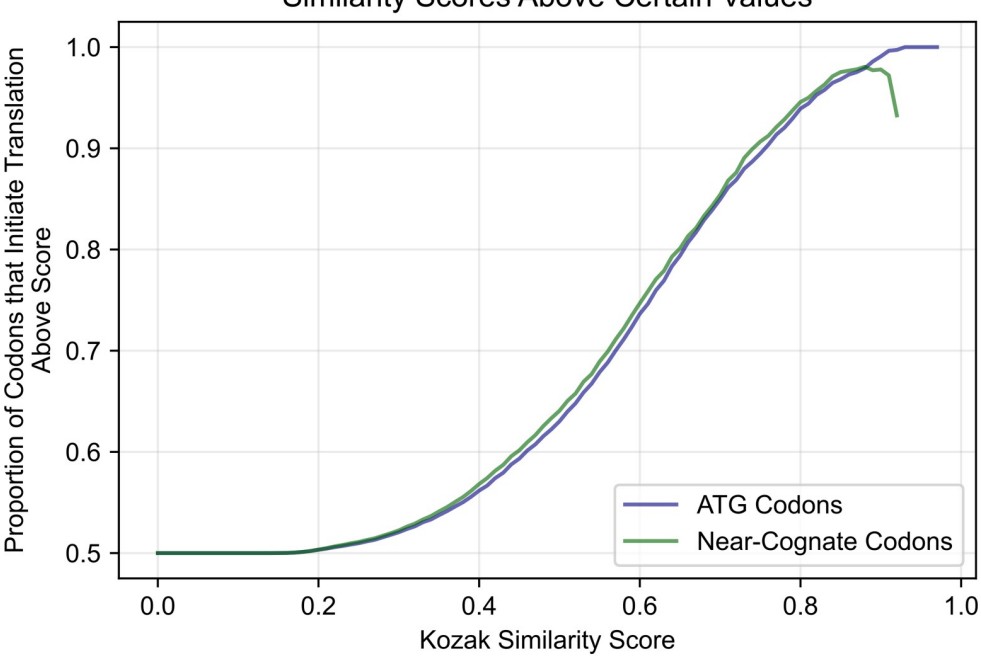

| Kozak Similarity Score | Proportion of ATGs that Initiate Translation above Kozak Similarity Score | Proportion of Near-Cognate Codons that Initiate Translation Above Kozak Similarity Score |
|---|---|---|
| **0.00** | 0.5000 | 0.5000 |
| **0.05** | 0.5000 | 0.5000 |
| **0.10** | 0.5000 | 0.5000 |
| **0.15** | 0.5001 | 0.5000 |
| **0.20** | 0.5033 | 0.5035 |
| **0.25** | 0.5098 | 0.5110 |
| **0.30** | 0.5206 | 0.5224 |
| **0.35** | 0.5375 | 0.5412 |
| **0.40** | 0.5618 | 0.5683 |
| **0.45** | 0.5933 | 0.6015 |
| **0.50** | 0.6302 | 0.6406 |
| **0.55** | 0.6789 | 0.6893 |
| **0.60** | 0.7364 | 0.7469 |
| **0.65** | 0.7938 | 0.8011 |
| **0.70** | 0.8496 | 0.8541 |
| **0.75** | 0.8948 | 0.9067 |
| **0.80** | 0.9393 | 0.9458 |
| **0.85** | 0.9682 | 0.9753 |
| **0.90** | 0.9908 | 0.9779 |
| **0.95** | 1.0000 | No codons above score |

**Fig 6. Proportion of ATG and near-cognate codons that initiate translation with KSSs above certain values.** The graph and table were both generated from the same results, using balanced data., i.e., an equal background proportion of positives and negatives.

process, the averaging of many such trees reduces variance and helps avoid overfitting. With an overfit model, data inputs that vary slightly from trained data could have volatile classifications that are not reliable. The RFC, which implements the averaging process, may produce greater accuracy than any one decision tree alone [31].

Accordingly, an RFC was implemented as a separate algorithm to elucidate whether codons initiate translation. To create such an algorithm, the feature variables of codons for the RFC to be trained on were first assigned. For an ATG classifier, these variables designated the ten nucleotides that preceded the codon, and ten that followed it. This range was chosen as studies suggest that alterations of bases in some of these positions are highly impactful, and may define whether a flanked codon is an "optimal, strong, [or] moderate" translation initiation site [14–21]. Although secondary structures influence translation, the successful identification of feature patterns may require exceptionally large amounts of training data that are currently unavailable. Of note, the number of training samples required to differentiate data increases exponentially as the number of attributes in a model increases [32]. Since five features are needed to designate whether a nucleotide at each position, $n$, is either adenine, guanine, cytosine, thymine, or missing, $5^n$ distinct data (i.e., enough to cover all possible data variations) may be required for a model to best approximate the impact of each nucleotide for every position that is considered. By having our models trained on a relatively small number of nucleotides known to influence translation initiation, we sought to optimize predictive power with limited data. For a near-cognate codon classifier, we included additional features to designate the nucleotide in the first base position of the codons (i.e., the underlined: CTG, GTG, TTG) since the nucleotide at this position may significantly impact translation initiation from these codons [22, 23].

Using the package, imbalanced-learn, in Python, we created the RFC models [33]. The ATG RFC was trained using an imbalanced set of 12,603 ATG codons known to initiate translation (positives), and 3,433 of 34,097 generated distinct ATG codons that are believed not to initiate translation (negatives). The set of 3,433 negatives consisted of the total of 1,805 sequences that were not missing nucleotides, and 1,628 (i.e., ten percent fewer) randomly sampled negatives of the remaining 31,697 that were missing nucleotides. We left out five percent of the total 3,433 negatives used (172 ATGs that do not initiate translation), as well as the same number of positives (172 ATG translation initiation codons) from the training data to constitute our test dataset. In this way, accuracy would be based on unbiased data that was balanced with 344 combined cases of equally occurring positives and negatives.

The accuracy of the RFC model on the balanced 344 cases was 87.79%. In other words, the algorithm correctly categorized 302 of the 344 ATGs, based on the sequences flanking each codon. This accuracy is high in comparison to the 79.85% accuracy achieved using the KSS-based classifier. We also calculated the AUROC score of the model to be 0.95, which is high as well. Increasing the parameter value designating the total number of decision trees included in the RFC had no visible effect on model performance. Other parameters were also left unchanged for optimal predictions.

The same procedure was used to create an RFC for near-cognate codons as carried out for ATG codons, using data available for near-cognate codons. To prevent imbalanced data bias in the accuracy measurement for the near-cognate RFC, data that was equally representative of all near-cognate codons was set aside to form the test dataset. Since the model was trained on

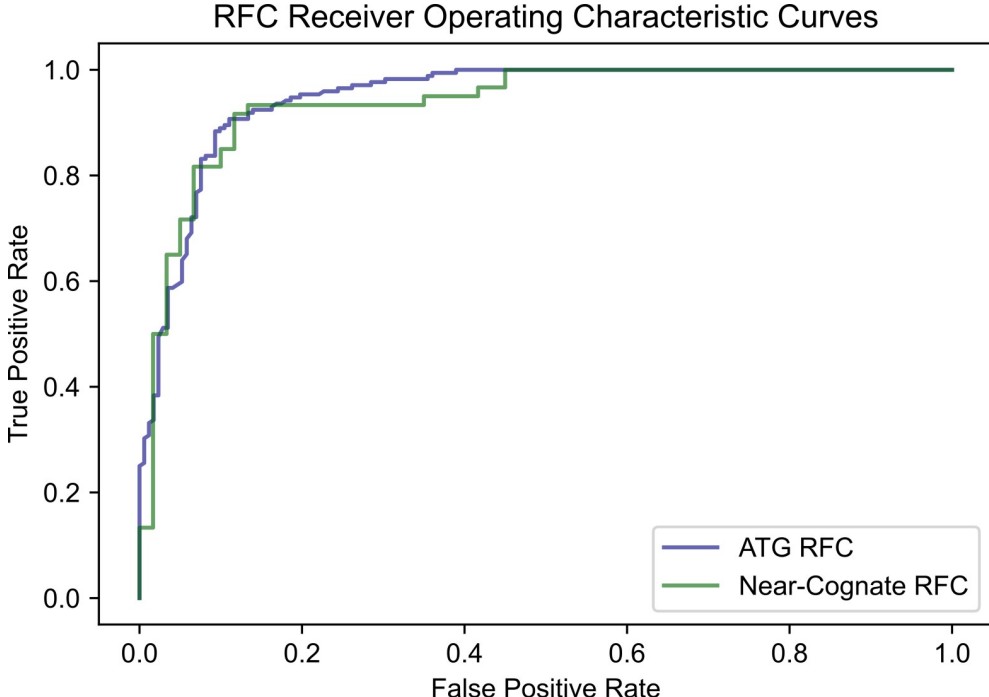

**Fig 7. ROC curves of the ATG and near-cognate random forest classifiers.** The AUROC score (area under the curve) of the ATG RFC is equal to 0.95. The AUROC score of the near-cognate RFC is equal to 0.94.

CTG, GTG, and TTG initiation codons, twenty positives and negatives were randomly isolated for each of these codons prior to training. When run on this separated, balanced set of 120 data points, the trained near-cognate RFC performed with 85.00% accuracy. The AUROC score of the near-cognate classifier was calculated to be 0.94. ROC curves for the ATG and near-cognate RFCs are shown in Fig 7.

## Analysis of the TITER neural network as a benchmark

There exist two other models for predicting both ATG and near-cognate translation initiation codons. The most recent is the TITER machine learning algorithm [12], which addresses limitations of the first model. We analyzed TITER as a benchmark to compare it with the performance of our models.

TITER is a deep learning-based framework that predicts whether a codon initiates translation based on the product of two calculations, which is termed TISScore. One constituent is based on the frequency of the codon of interest (e.g., ATG, CTG, GTG, etc.) in the dataset to initiate translation. The second involves the averaging of calculated scores for a codon with flanking sequences across thirty-two neural networks. A large number of neural networks was used as part of a bootstrapping technique to account for training data imbalance.

Although TITER has a high AUROC score of 0.89 [12], ROC curves can present an "overly optimistic" evaluation of a model's performance "if there is a large skew in the class distribution" [28, 29]. This assessment is based on the true positive and false positive rates of the model–and an imbalance of positives and negatives may distort its calculation [34]. One questions whether the test sample of the model is skewed as it consists of 767 positive and 9,914 negative samples in total [12]. Although the authors noted special procedures to account for the data imbalance of the training dataset, it is not clear if such procedures were used for the test dataset.

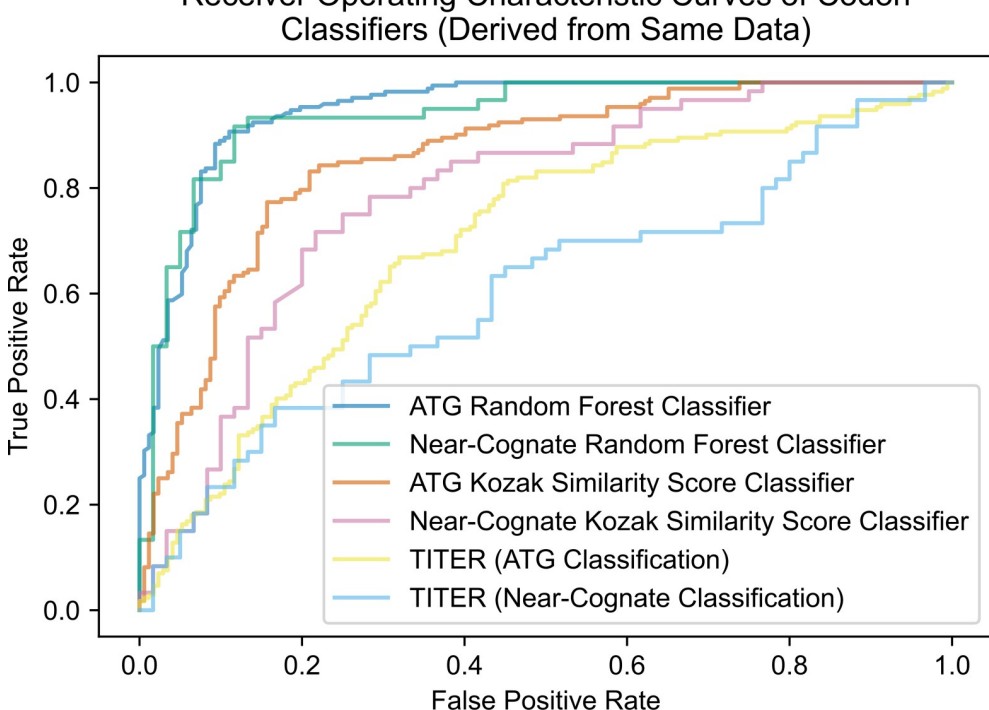

**Fig 8. ROC curves of all ATG and near-cognate classifiers derived from same test data.** All classifiers were run on the ATG and Near-cognate RFC test datasets, and their ROC curves were superimposed. The AUROC scores of the ATG and near-cognate RFCs are 0.95 and 0.94, respectively. The AUROC scores of the ATG and near-cognate KSS classifiers are 0.86 and 0.79 respectively on these test datasets. TITER's AUROC scores are 0.62 and 0.60 for ATG and near-cognate codons, respectively.

Since TITER was open-source, TITER's accuracy was averaged across a hundred balanced subsets from its test dataset. Using all 767 positive samples, 767 negatives were randomly sampled from the 9,914 total negatives, across the hundred runs to account for the data imbalance. Through this technique, the unbiased average of the model accuracy was calculated to be 66.94%. This was the accuracy achieved by the best cutoff, 0.5, of the TISScore for classification. When run on the same sequences comprising the RFC test datasets (with sequences extended to include the additional features TITER was trained with), TITER demonstrated 62.21% and 58.33% accuracy for ATG and near-cognate codons, respectively. These values were lower than the 75.60% and 79.85% accuracy achieved using the KSS scoring system for ATG or near-cognate codons, or the 85.00% and 87.79% accuracy achieved using RFC models. The fact that TITER was trained with less data than the RFC models presented here could account for a reduced predictive power. Specifically, it was generated using 9,776 positive samples and 94,899 negatives compared to the total 15,016 positives and 175,168 negatives used for the RFCs.

The performance of TITER may also be a result of the large number of features that this machine learning model incorporated. Although contemporary research suggests a few bases that flank a codon greatly influence translation initiation from this site [14–21], TITER analyzes a total of two hundred bases that flank each codon. Compared to our approach of analyzing twenty nucleotides that flank the codon, TITER may implement up to 180*5 = 900 additional features via its one-hot encoding procedure. The expression '180*5' is used because any one base at the 180 extra positions is represented by five features to designate whether the

base is adenine, guanine, cytosine, tyrosine, or is missing. Although the TITER publication mentions feature reduction in the hidden layer of the neural networks, it is not clear how much feature reduction occurred and whether features with significant correlations were inadvertently reduced. Of note, an excess of features may decrease effectiveness in machine learning because the number of training samples required to differentiate the data increases exponentially as the number of attributes in a model increases. Thus, predictive power is lost. In fact, this phenomenon is termed the "curse of dimensionality" in Data Science [32].

There are a number of reasons that our model's performance may improve predictions of translation initiation codons: a) feature reduction, b) implementation of the random forest classifier, which is more robust to outliers and erroneous instances (especially when data is limited), c) creation of two models to account for properties of different data types (i.e., ATG codons versus near-cognate codons), d) use of sampling *without* replacement, which preserves natural variations found in data (in place of bootstrapping).

ROC curves for TITER, our RFCs, and the KSS classifiers derived from the same dataset are superimposed in Fig 8.

## All else equal, RFCs perform better than neural networks

To demonstrate that the RFC has superior performance to the neural network with all else equal, we evaluated performance of optimized convolutional neural networks using identical data and features as those discussed in the RFC section. The best ATG neural network yielded about 84.59% accuracy and an AUROC score of 0.91. The best near-cognate neural network had about 76.67 accuracy and an AUROC score of 0.90. These results indicate that RFCs perform better than neural networks for our task. We speculate the better performance may be due to neural network overfitting, especially for the more variable near-cognate data. Lower neural network accuracy is consistent with other studies when only limited data are available, i.e., when data may not number in the millions [35, 36].

## Model selection and integration into software

Of the models generated, the RFCs appeared the best choice to use for predicting translation initiation sites. Thus, we used the RFCs trained with high accuracy at 87.8% for ATGs and 85.0% for near-cognate codons. The RFCs outperform other published models designed for the same function, for example, exceeding TITER by more than 18% in accuracy.

As a next step, we used the RFCs to identify repeat-length-independent (RLI) translation initiation associated with neurologic diseases. To carry this out, the RFC models were implemented into software. Developed in Python, the program could be used to evaluate a total sequence consisting of the upstream region followed by ten nucleotide sequence repeats to represent the repeat expansion. Ten sequence repeats may be adequate to capture the repeat expansion effect on translation initiation from upstream codons as well as codons within the repeat expansion itself since ten nucleotide sequence repeats are at minimum thirty bases long, and the integrated model only uses the ten bases that flank each side of a codon for analysis. Nucleotides within this range have been shown to strongly impact translation initiation [14–21].

The model can scan through each codon in the sequence and return a prediction from the implemented RFCs. If a codon encountered is 'ATG,' then the ATG RFC with 87.79% accuracy predicts whether it initiates translation based on the ten sequences flanking each side of the codon. Otherwise, if the codon encountered is a near-cognate codon, then the near-cognate RFC with 85.00% accuracy predicts whether it initiates translation via the same procedure. Next, the program virtually simulates translation from each predicted codon and filters out

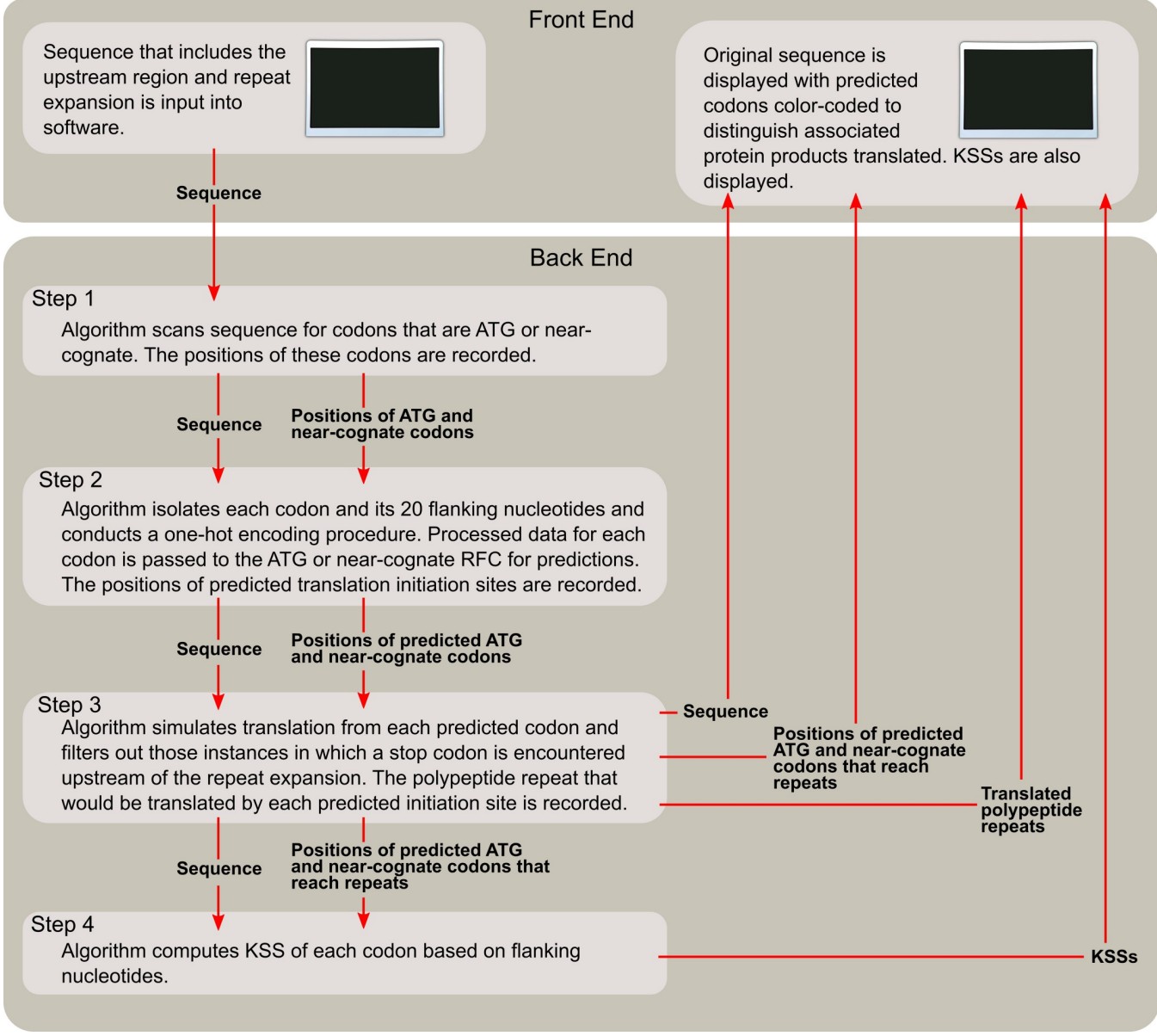

**Fig 9. Software architecture.**

those instances in which a stop codon (TAG, TGA, or TAA) is encountered upstream of the repeat expansion. This feature was implemented to remove codons from consideration if their initiated translation would not reach the repeat expansion and produce the repeat proteins that may be associated with neurologic disease. The program can determine the nucleotide sequence repeat as well as the associated translation product from the remaining codons. Finally, the program outputs a visualization of the input sequence, with predicted codons color-coded to distinguish the associated product translated. Software architecture is depicted in Fig 9.

In the figures that follow, nucleotides have a bold font to distinguish initiation codons that the software models were trained on. These codons include the canonical start codon ATG as well as near-cognate codons CTG, GTG, and TTG. Because the features of the three near-

TACCTTGTAGAAAGCGCC**ATTG**GAGCCCCGCACTTCCACCACCAGCTCCTCCATCTTCTCTTCAGC
C**CTG**CTAGCGCCGGGAGCCCGCCCCCGAGAG**GTG**GG**CTG**CGGGCGCTCGAGGCCC

**Fig 10. An example of the formatting scheme in software output.** This example shows predicted codons that are color-coded based on their reading frame: 'ATT,' 'TTG,' 'CTG,' 'AGG,' 'GTG,' and 'CTG.' Codons that the models were trained on show up with bold formatting. If there is an overlap between predicted initiation codons (i.e., one or two nucleotides overlap between predicted codons), the color of the overlapped region is the same as the color of the next predicted codon.

cognate codons were used to extrapolate classifications of the other, less researched near-cognate codons (viz., AAG, AGG, ACG, ATC, ATT, and ATA), it is possible to incur false predictions for these less studied instances. Thus, these six near-cognate codons are designated only with color-coding without bolding to denote that they should be acknowledged with less confidence. If there is an overlap between predicted initiation codons (i.e., one or two nucleotides overlap between predicted codons), the color of the overlapped region is the same as that of the next predicted codon to prevent confusion. The overlapped region may or may not be bolded depending on whether the software was trained on this next codon. An example of formatted output is shown in Fig 10. We also output the KSSs of each predicted codon to two decimal points, as the score could be a useful metric to evaluate translation initiation likelihood. This may be approximated through comparison of KSSs of a codon to the reference table and graph (Fig 6). In this way, KSS might be used to further identify significant predictions.

## Software identification of known RLI translation initiation sites

After the software was completed, its ability to distinguish RLI translation initiation sites was analyzed. We first identified translation initiation codons upstream of repeats in the following genes in which RAN translation is known to occur: *C9orf72* (associated with amyotrophic lateral sclerosis and frontotemporal dementia), *FMR1* (associated with fragile X and fragile X-associated tremor/ataxia syndrome), *DM1* (associated with myotonic dystrophy type 1), and *HDL2* (associated with Huntington disease-like 2) genes. These examples were used as references for software performance. It should be noted that translation initiation codons identified for DM1 were obtained from an experiment that implemented a version of the conventional DM1 antisense strand that had been slightly modified to determine whether changes in its sequence could induce translation initiation from particular codons [37]. The associated upstream regions and repeat expansion sequences for each gene, as recorded in the National Center for Biotechnology Information database, were input into the software. Predictions were generated in order to determine whether nucleotide sequences corresponded to experimentally confirmed translation initiation codons (Table 1).

Comparison between the predictions and experimentally identified translation initiation codons demonstrated high performance of the software. In fact, all translation initiation sites previously identified from publications were correctly identified by the RFCs with only one exception: ATC, which was experimentally determined to initiate translation in the modified *DM1* antisense strand seven bases upstream of the repeat [37]. Importantly, the near-cognate RFC model successfully predicted all other instances of translation initiation from less researched near-cognate codons. This accuracy is surprising considering that the near-cognate RFC model was only trained on instances of CTG, GTG and TTG translation. As there was insufficient data to train the model on less used near-cognate codons (ATA, ATC, ATT, AGG, ACG, and AAG), predictions for these codons were extrapolated based on recognized patterns from CTG, GTG, and TTG examples. However, for the same reason that they were not

**Table 1. Previously published repeat length-independent translation initiation sites.**

| Gene | Codon | Number of Bases Upstream of Repeat | Peptide Repeat Translated | Kozak Similarity Score |
|---|---|---|---|---|
| *C9orf72* (Sense) [4] | AGG | 1 | Poly-GR | 0.66 |
| | CTG | 24 | Poly-GA | 0.69 |
| *C9orf72* (Antisense) [4] | ATG | 194 | Poly-PG | 0.61 |
| *FMR1* (Sense) [10, 38] | GTG | 11 | Poly-G | 0.70 |
| | ACG | 35 | Poly-G | 0.80 |
| | ACG | 60 | Poly-G | 0.71 |
| *DM1* (Antisense)* [37] | ATC | 7 | Poly-A | 0.61 |
| | ATG | 17 | Poly-S | 0.66 |
| | ATT | 23 | Poly-S | 0.74 |
| *HDL2* (Antisense) [37] | ATC | 6 | Poly-Q | 0.74 |

* The DM1 antisense strand had a slightly modified sequence [37].

included in model training, near-cognate codons that are not CTG, GTG, or TTG should be acknowledged with less confidence in predictions out of concern they may be false positives.

## Predicted translation initiation sites associated with neurologic diseases

As shown above, experimentally identified translation initiation codons for *C9orf72*, *FMR1*, *DM1*, and *HDL2* were confirmed by the model presented here (Table 1, Figs 11 and 12). The software was then used to predict translation initiation codons associated with repeats in genes that cause neurologic diseases that have not been experimentally identified. These include additional genes *HTT*, and *DM2* (Fig 13). Best predictions for the translation initiation codons for genes are highlighted in yellow in Figs 11–13 based on rules we suggest in the next section. Predicted translation initiation codons with relatively high KSSs were flagged, alongside RFC probabilities for the analyzed genes (Table 2). The RFC probability for a codon is calculated as the number of decision trees in the classifier that predict translation initiation to occur, divided by the total number of trees used, viz., one thousand in our case. A positive correlation between KSSs and RFC probabilities is observed, particularly for ATG codons. One advantage of the KSS is that it is an entirely separate scoring system from the RFCs. In all cases, predicted translation initiation sites are not shown if they have a downstream stop codon located in the same reading frame before the repeat.

Results displayed in the figures and table indicate translation initiation sites for proteins translated from the repeat. Of note, the average KSS of all upstream predicted codons is about 0.66. With reference to the table in Fig 6, approximately 80% of ATG and near-cognate codons with a score above 0.65 are estimated to initiate translation from a background population of equally occurring translation initiation codons (positives) and codons believed not to initiate translation (negatives).

With respect to the sequence upstream of the repeat on the *C9orf72* sense strand, the software predicts a codon to initiate translation of poly-GA, and another to initiate translation of poly-GR—both of which have been confirmed through experimentation [4]. In the antisense strand, there are ten codons that could initiate translation of poly-PR, and six predicted with respect to poly-PG. The ATG located 194 bases upstream of the repeat expansion has been confirmed [4].

Predictions for translation initiation codons from the *FMR1* sense strand upstream from the repeat identify nine codons that could be used to initiate translation of poly-G, and two for poly-R. The translation initiation codon GTG located 11 bases upstream, the ACG located 35

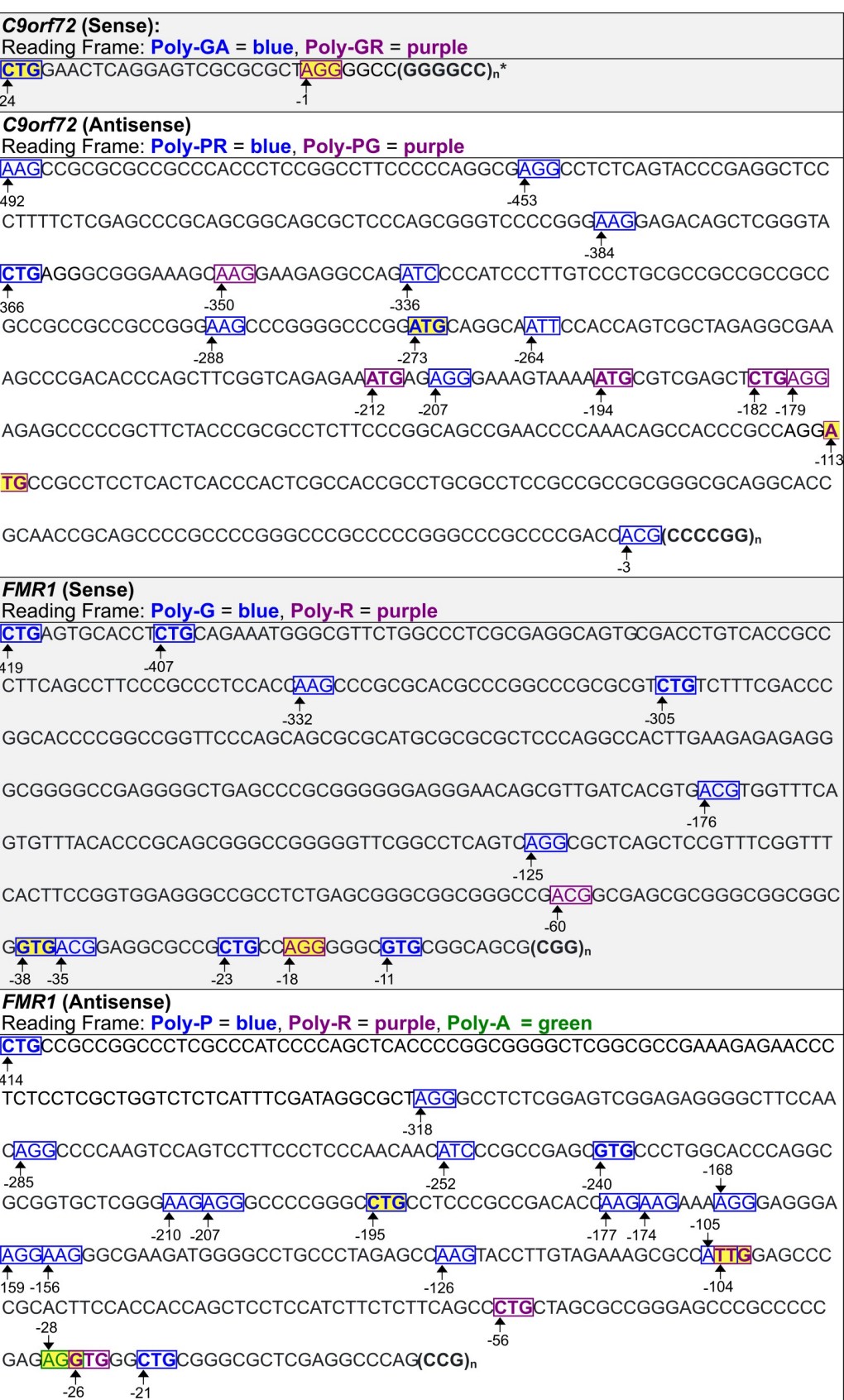

**Fig 11. Predicted translation initiation codons for C9orf72 *and* FMR1.** Best predictions per reading frame are highlighted in yellow. Predicted codons that the models were trained on show up with bold formatting. The numbers in this Table and subsequent ones indicate the position of the bases upstream of the repeat. * A predicted translation initiation codon is located 1 base upstream of the repeat, and overlaps with an AGG which is partly in the repeat.

bases upstream, and ACG located 60 bases upstream have been confirmed experimentally [38]. The antisense upstream region has a total of sixteen codons predicted to initiate translation of poly-P, three for poly-R, and one for poly-A.

For the *DM1* sense strand upstream from the repeat, the software predicts three codons that initiate translation of poly-C, and two that initiate translation of poly-A. Interestingly, every CTG within the CTG repeat expansion is predicted to initiate translation of poly-L; however, only the first has a relatively high KSS (0.67). Predictions for the DM1 antisense strand are different from those produced for the experimentally modified DM1 antisense strand (Table 1). Namely, there is no predicted ATG located 17 bases upstream of the repeat expansion, nor a predicted ATT located 23 bases upstream of the repeat expansion because sequences that border the predicted codons in the modified strand differ from those bordering the same codons in the unmodified version. In the unmodified antisense strand, there are seven codons predicted to initiate poly-A translation, and one to initiate translation of poly-S. Also, there are no predicted translation initiation codons in the reading frame of poly-Q which suggests that this polypeptide might be initiated from the repeat expansion, possibly by repeat length-dependent folding.

With respect to the *HDL2* sense strand upstream from the repeat, the software predicts seven codons to initiate translation of poly-L, one to initiate translation of poly-C, and two to initiate translation of poly-A. Furthermore, the software suggests that every CTG of the CTG repeat expansion, aside from the first one in the sense strand, can initiate translation of poly-L, presumably because of the flanking nucleotides. In the antisense strand, there are seventeen codons predicted to initiate translation of poly-Q, three for poly-S, and two for poly-A. The predicted ATC located 6 bases upstream of the repeat expansion in the antisense strand has been confirmed [37].

Predictions for translation initiation codons from the *HTT* sense strand upstream from the repeat identify seventeen codons that initiate translation of poly-Q, and four for poly-A. From the antisense upstream region, sixteen codons are predicted to initiate translation of poly-L, and nine for poly-A. The software also suggests that every CTG of the CTG repeat expansion, aside from the first one in the antisense strand can initiate translation of poly-L.

Predictions for the *DM2* sense strand upstream from the repeat identify five codons used for translation initiation of poly-PACL, two for poly-CLPA, and three for poly-LPAC. Moreover, the software predicts the first two CTGs of the CCTG repeat expansion to initiate translation of poly-LPAC. In the antisense strand, there are three codons predicted to initiate poly-RQAG translation, five to initiate translation of poly-GRQA, and one to initiate translation of poly-QAGR.

## Identifying best predicted initiation codons

To choose the best predictions for initiating codons, we constructed a reference table that places predicted codons into five categories (Fig 14). The categories are sorted in accordance with the amount of data used for each codon in model training, the model's accuracy, and trends noted with KSS. We take into consideration ATG's greater tendency to initiate translation relative to other codons. Codons other than ATG, CTG, GTG, or TTG are placed in a lower category because they are extrapolations (were not used in model training). We suggest

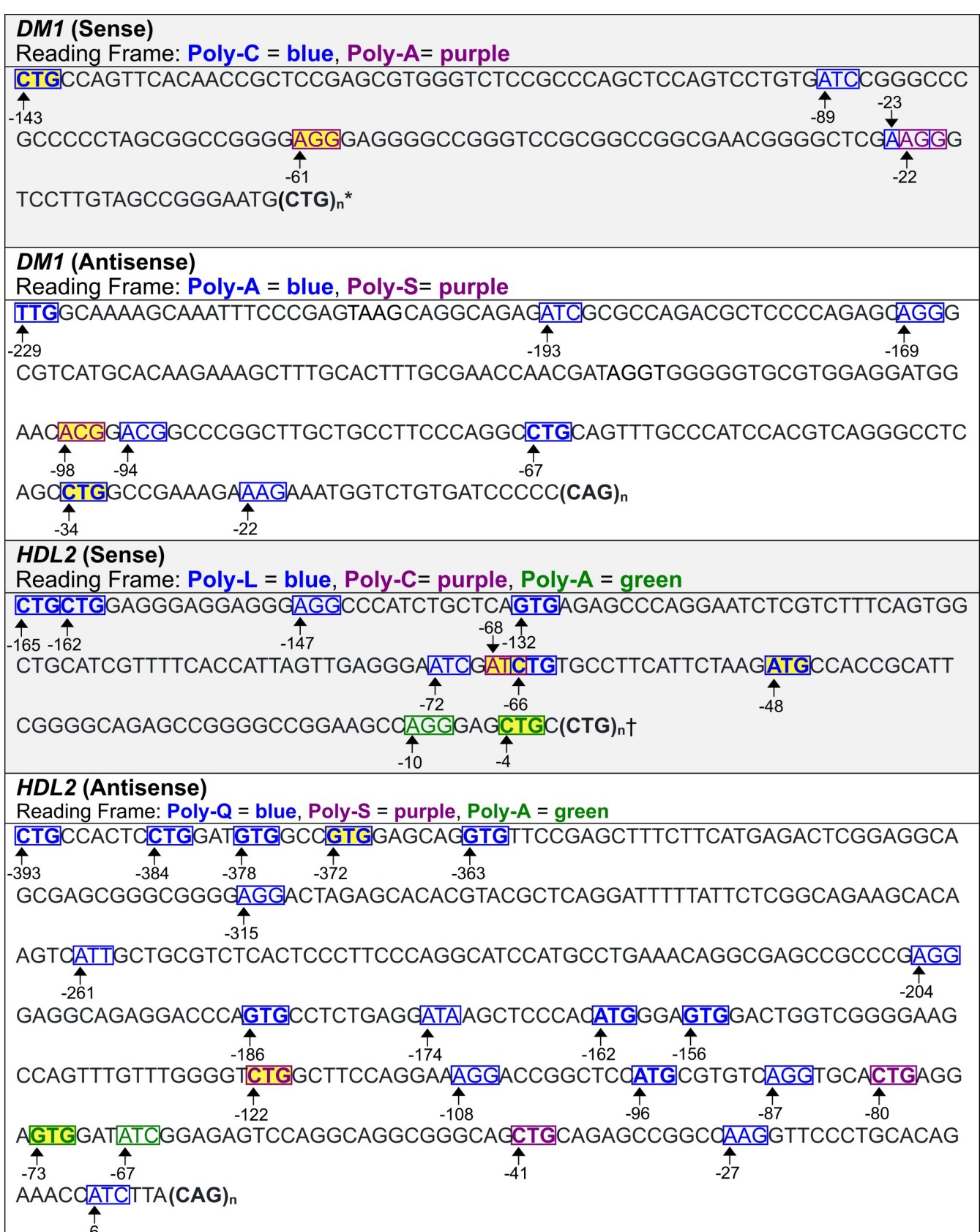

**Fig 12. Predicted translation initiation codons for DM1 *and* HDL2.** Predicted codons that the models were trained on show up with bold formatting. Best predictions per reading frame are highlighted in yellow. * Every CTG within the repeat could initiate translation of poly-L. The CTG at position 0 may be the best predicted initiation codon as its flanking sequence is closest to the KCS. † Every CTG within the repeat, aside from the first one, is predicted to initiate translation, presumably because of the flanking nucleotides.

sorting predictions based on a 'Prediction Categories' reference table, with 1 most preferred and 5 least preferred. Predictions in the same category are to be sorted by their KSS or RFC probability. One advantage of the KSS is that it is an entirely separate scoring system from the RFCs.

## Materials and methods

### Data acquisition

Examples of translation initiation were primarily obtained from ribosome profiling, mass spectroscopy, and CRISPR-based techniques across different human cell types and under different conditions [39]. These data include sequences of 12,094 examples of translation initiated from ATG, as well as 2,180 examples of translation initiated from near-cognate codons. Translation initiation sites were also captured by quantitative translation initiation sequencing of genes in cultured human kidney cells [40]. Their annotated sequences were collected from the Ensembl gene annotation system (version 84) [41]. These methods procured 509 and 203 more examples of ATG and near-cognate initiation codons, respectively. In all, we collected 12,603 instances of translation initiation from ATG, and 2,413 instances of translation initiation from near-cognate codons to use in this study.

To obtain examples in which translation does not initiate from ATG (negatives), we used the same transcripts from which positives were derived and recorded nucleotides that flanked ATG codons. Then, we eliminated all instances in which flanking sequences matched any of the 12,603 sequences bordering the known ATG translation initiation sites, leaving 34,097 negatives. We repeated the same procedure to identify negatives for near-cognate codons that do not initiate translation. We found examples of CTG, GTG, and TTG codons in which flanking sequences did not match any of that of the known near-cognate initiation codons, leaving 141,071 negatives.

### Random sampling

All random sampling was conducted without replacement. This method is preferred for KSS evaluations of ATG and near-cognate codons, as the precision of population estimates is higher than that produced by sampling with replacement [42]. Furthermore, sampling without replacement to generate training datasets introduces greater variation for model training.

### Random forest classifiers

Using the open-source package, imbalanced-learn, in Python, we created the RFC models [33]. The ATG RFC was trained on an imbalanced set of 12,432 ATG codons known to initiate translation (positives), and 3,261 ATG codons that are believed not to initiate translation (negatives). The set of 3,261 negatives consisted of 1,716 sequences that were not missing nucleotides, and 1,545 (ten percent fewer) randomly sampled negatives of the remaining 31,697 that were missing nucleotides. To clarify, missing nucleotides are registered in the case that a recorded codon is located exceedingly close to the 5' or 3' end of an mRNA construct. In such a circumstance, there may not be a full ten bases both preceding and following the codon. The sampling technique was performed to slightly offset the proportion of negatives with and

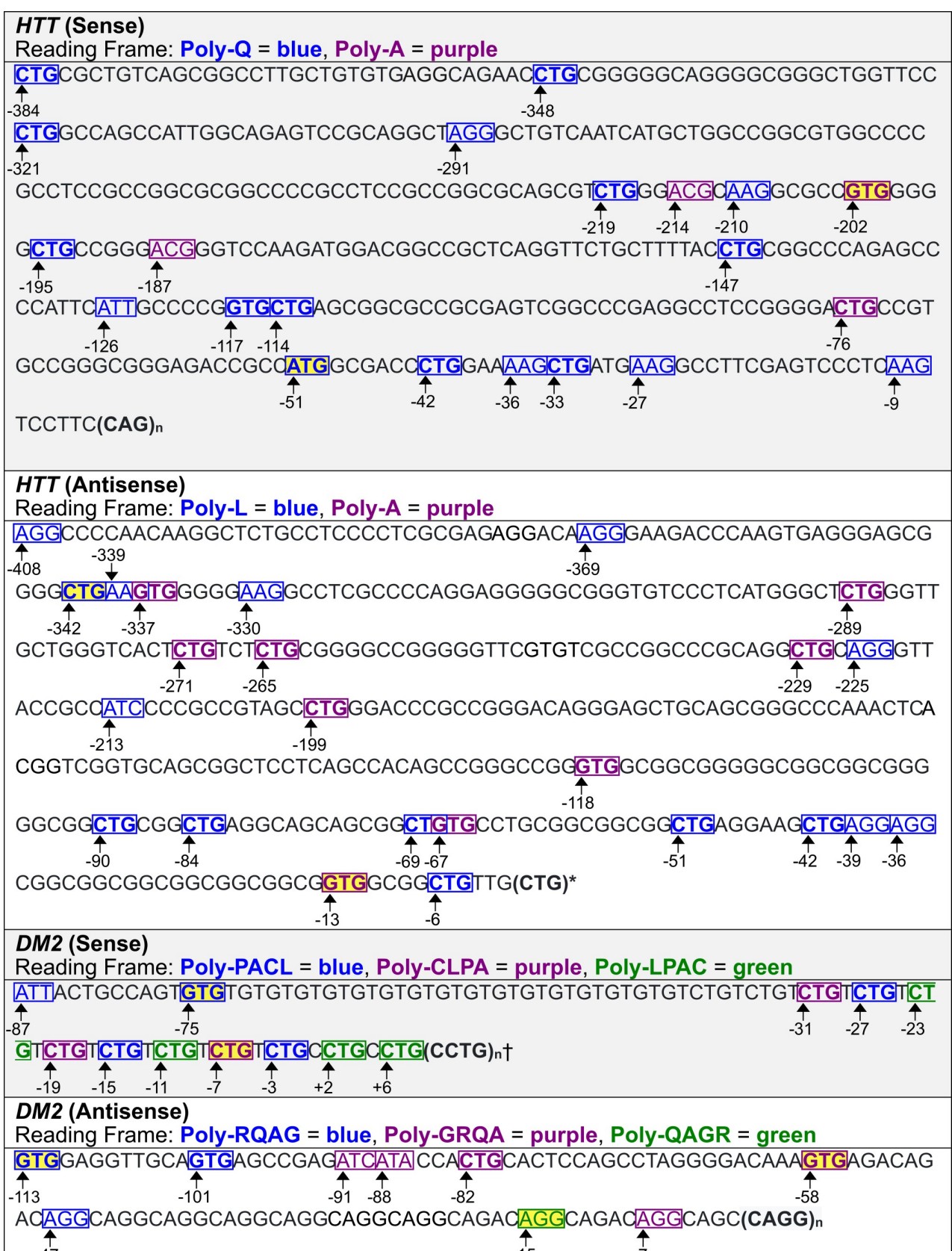

**Fig 13. Predicted translation initiation codons for *HTT* and *DM2*.** Predicted codons that the models were trained on show up with bold formatting. Best predictions per reading frame are highlighted in yellow. * Every CTG within the repeat, aside from the first one, is predicted to initiate translation, presumably because of the flanking nucleotides. † Two predicted translation initiation codons are within the repeat. A CTG within the repeat at position 1 could be the best predicted initiation codon in the poly-LPAC reading frame based on rules suggested in Fig 14.

without missing bases in the opposite direction. In this way, more negatives without missing bases would be used for model training. Using the original imbalanced set of negatives, with the majority missing bases, would cause the model to inaccurately assess the effect of missing nucleotides on a codon's ability to initiate translation. Furthermore, using a slightly larger proportion of negatives that had a complete sequence profile resulted in improved accuracy for distinguishing codons that were not missing nucleotides. This method is useful since sequences are less often encountered with missing nucleotides in real-world applications.

To account for the imbalance of positives and negatives, the RFC had decision trees generated from 3,576 negatives, and the same number of randomly sampled positives. One thousand such trees were used, since this number is generally recommended as a starting point for the generation of an RFC [43]. Of the total number of features, $n$, a total of $\sqrt{n}$ features were considered at each branch of the trees to best classify data. Using too many or too few features could have prevented the model from recognizing the best indicators for classification [43]. Each decision tree also had the requirement of grouping at least two codon instances to a certain classification. This constraint reduced the risk of overfitting, yet still allowed tree capacity to differentiate between subtly differing codons. Thus, the trees could better identify precise feature patterns to associate with a particular classification, and remain reliable in face of new, unencountered data.

We evaluated the accuracy of the RFC model with the above configurations. Parameters such as the minimum number of codons to group for classification could then be adjusted to improve predictive power, as necessary. However, parameters were best left unchanged for optimal predictions. To create a separate classifier for near-cognate codons, we repeated the same procedures to create an RFC for near-cognate codons as we had carried out for ATG codons, this time using data available for all near-cognate codons. RFC design is depicted in Fig 15.

## Convolutional neural networks

Convolutional neural networks were developed in Python using TensorFlow and Keras [44, 45] using the same datasets as the RFCs to which they were compared. For the ATG model, we reduced 115 features into a 70-dimensional space via convolution layers and max pooling. Three additional hidden layers were used containing 16, 8, and 4 nodes respectively. To account for the data imbalance, the minority class was upsampled to match the majority class. For a near-cognate neural network, we optimized performance using the same model structure as used for the ATG model, with the addition of batch normalization.

## Discussion

As shown here, RFCs were able to successfully predict most translation initiation codons associated with neurologic repeat expansion diseases that have been experimentally identified. The same models also predicted other translation initiation codons of repeat expansions for neurologic diseases that have not been experimentally identified. Of note, the software predicted translation initiation sites with more than 18% accuracy than the TITER neural network.

Regardless of the quality of a model, its predictions should not be interpreted as evidence. Instead, predictions should be recognized as likely possibilities that warrant further

**Table 2. Translation initiation codons with high Kozak similarity scores\*.**

| Gene | Codon | Number of Bases Upstream of Repeat | RFC Probability | Kozak Similarity Score | Translated Polypeptide Repeat |
|---|---|---|---|---|---|
| *C9orf72* (Sense) | CTG† | 24 | 0.69 | 0.66 | Poly-GA |
| | AGG | 1 | 0.55 | 0.69 | Poly-GR |
| *C9orf72* (Antisense) | **ATG‡** | 113 | 0.82 | 0.75 | Poly-PG |
| | AAG | 350 | 0.64 | 0.84 | Poly-PG |
| | ACG | 3 | 0.6 | 0.79 | Poly-PR |
| | AAG | 288 | 0.63 | 0.73 | Poly-PR |
| | AAG | 384 | 0.69 | 0.77 | Poly-PR |
| *FMR1* (Sense) | AGG | 18 | 0.7 | 0.83 | Poly-R |
| | ACG | 60 | 0.66 | 0.71 | Poly-R |
| | ACG | 35 | 0.8 | 0.79 | Poly-G |
| | **GTG** | 38 | 0.77 | 0.76 | Poly-G |
| | AAG | 332 | 0.57 | 0.83 | Poly-G |
| *FMR1* (Antisense) | AGG | 28 | 0.62 | 0.71 | Poly-A |
| | **GTG** | 26 | 0.59 | 0.73 | Poly-R |
| | **CTG** | 56 | 0.54 | 0.7 | Poly-R |
| | ATT | 105 | 0.66 | 0.81 | Poly-P |
| | AAG | 156 | 0.58 | 0.78 | Poly-P |
| | AAG | 177 | 0.64 | 0.85 | Poly-P |
| | **CTG** | 195 | 0.73 | 0.74 | Poly-P |
| | AGG | 207 | 0.68 | 0.84 | Poly-P |
| | ATC | 252 | 0.55 | 0.8 | Poly-P |
| | AGG | 285 | 0.54 | 0.77 | Poly-P |
| | AGG | 318 | 0.71 | 0.74 | Poly-P |
| *DM1* (Sense) | AAG | 23 | 0.52 | 0.62 | Poly-C |
| | AGG | 61 | 0.61 | 0.77 | Poly-A |
| | **CTG** | 0 | 0.67 | 0.67 | Poly-L |
| *DM1* (Antisense) | **CTG** | 34 | 0.74 | 0.87 | Poly-A |
| | AGG | 169 | 0.54 | 0.85 | Poly-A |
| | ATC | 193 | 0.69 | 0.81 | Poly-A |
| | ACG | 98 | 0.67 | 0.86 | Poly-S |
| *HDL2* (Sense) | ATC | 72 | 0.58 | 0.71 | Poly-L |
| | ATC | 68 | 0.56 | 0.52 | Poly-C |
| | AGG | 10 | 0.72 | 0.84 | Poly-A |
| *HDL2* (Antisense) | ATC | 6 | 0.51 | 0.74 | Poly-Q |
| | AAG | 27 | 0.63 | 0.8 | Poly-Q |
| | ATT | 261 | 0.62 | 0.81 | Poly-Q |
| | **GTG** | 372 | 0.71 | 0.83 | Poly-Q |
| | **GTG** | 378 | 0.6 | 0.71 | Poly-Q |
| | **CTG** | 122 | 0.7 | 0.68 | Poly-S |
| | ATC | 67 | 0.59 | 0.69 | Poly-A |
| *HTT* (Sense) | AAG | 27 | 0.63 | 0.76 | Poly-Q |
| | **CTG** | 33 | 0.63 | 0.72 | Poly-Q |
| | **CTG** | 42 | 0.85 | 0.87 | Poly-Q |
| | **ATG** | 51 | 0.94 | 0.89 | Poly-Q |
| | AAG | 210 | 0.58 | 0.72 | Poly-Q |
| | **CTG** | 348 | 0.65 | 0.74 | Poly-Q |
| | ACG | 187 | 0.59 | 0.75 | Poly-A |
| | **GTG** | 202 | 0.74 | 0.85 | Poly-A |

(*Continued*)

**Table 2.** (Continued)

| Gene | Codon | Number of Bases Upstream of Repeat | RFC Probability | Kozak Similarity Score | Translated Polypeptide Repeat |
|---|---|---|---|---|---|
| *HTT* (Antisense) | ATC | 213 | 0.66 | 0.76 | Poly-L |
| | AGG | 225 | 0.54 | 0.7 | Poly-L |
| | AAG | 330 | 0.69 | 0.73 | Poly-L |
| | **CTG** | 342 | 0.72 | 0.7 | Poly-L |
| | AGG | 369 | 0.64 | 0.76 | Poly-L |
| | **GTG** | 13 | 0.83 | 0.84 | Poly-A |
| | **GTG** | 118 | 0.64 | 0.72 | Poly-A |
| | **CTG** | 199 | 0.73 | 0.81 | Poly-A |
| | **CTG** | 229 | 0.71 | 0.71 | Poly-A |
| | **CTG** | 271 | 0.69 | 0.71 | Poly-A |
| | **GTG** | 337 | 0.61 | 0.73 | Poly-A |
| *DM2* (Sense) | **CTG** | 7 | 0.56 | 0.5 | Poly-CLPA |
| | **CTG** | -5 | 0.61 | 0.61 | Poly-LPAC |
| | ATT | 87 | 0.55 | 0.66 | Poly-PACL |
| *DM2* (Antisense) | AGG | 7 | 0.53 | 0.72 | Poly-GRQA |
| | **GTG** | 58 | 0.56 | 0.7 | Poly-GRQA |
| | ATA | 88 | 0.59 | 0.75 | Poly-GRQA |
| | AGG | 47 | 0.53 | 0.71 | Poly-RQAG |
| | AGG | 113 | 0.61 | 0.74 | Poly-RQAG |
| | AGG | 15 | 0.52 | 0.72 | Poly-QAGR |

* All codons that do not encounter a stop codon before the repeats are assessed. Predicted codons with a KSS above 0.70 are displayed. If no KSS within a reading frame is above 0.70, then the codon with the highest KSS is displayed–as in the case of the *C9orf72* sense strand.

† Highlighted codons represent most likely predictions per reading frame based on rules suggested in Fig 14. These codons are also highlighted in Figs 10–12. If no codons are highlighted in a reading frame, then the KSS of the best predicted codon is below 0.70.

‡ Bolded codons represent codons that the RFCs were trained on.

investigation. However, the significance of the algorithm's identification of translation initiation codons should not be understated. For example, these data may be important to use to guide treatment of these repeat diseases.

Although the machine learning models show promise in understanding of the pathogenesis of repeat expansion neurologic disorders, their use may be extended to other applications as well. For example, they may be used to predict the translation initiation codons (including alternative initiation codons) for genes that are not involved in repeat expansion disorders. One benefit of this implementation includes the ability to speculate protein products from a nucleotide sequence quickly and easily and without laboratory procedures. In order to accelerate the use of the RFCs, a version of the machine learning software that can predict translation initiation codons in any provided sequence is available at www.tispredictor.com/tis.

## Limitations

Unaccounted factors may limit the accuracy of our predictions in real-world applications. Whereas we examine consensus sequences from the NCBI database, there exist variations in the regions within repeats and bordering repeats. For example, there exist at least three transcript variants of C9orf72 with repeats that range from the tens to thousands in afflicted individuals [46]. It is important to note that: a) these repeats may contain intervening nucleotides that could introduce additional initiation codons and reading frames, b) these intervening

| Preference | Prediction Category |
|------------|---------------------|
| **1.** | ATG; KSS > 0.7 |
| **2.** | CTG, GTG, or TTG; KSS > 0.7 |
| **3.** | ATG; KSS < 0.7 |
| **4.** | CTG, GTG, or TTG; KSS < 0.7 |
| **5.** | Other |

Example:

**C9orf72 Antisense Prediction Rankings:**

| | |
|---|---|
| **1.** ATG at position -113 (KSS = 0.75, Prob = 0.82) | *Category 1* |
| **2.** ATG at position -194 (KSS = 0.61, Prob = 0.63) | |
| **3.** ATG at position -273 (KSS = 0.57, Prob = 0.60) | *Category 3* |
| **4.** ATG at position -212 (KSS = 0.61, Prob = 0.51) | |
| **5.** CTG at position -366 (KSS = 0.63, Prob = 0.56) | |
| **6.** CTG at position -182 (KSS = 0.58, Prob = 0.58) | *Category 4* |
| **7.** AAG at position -350 (KSS = 0.84, Prob = 0.65) | *Category 5* |

**Fig 14. Proposed order of best predictions.** We suggest categorizing predictions based on the above table, and then ordering them by category (1–5, with 1 most preferred). Predictions in the same category are to be sorted by either KSS or RFC probability. We show the ranking of C9orf72 antisense predictions as an example.

nucleotides may vary in different neural cells within the same patient. The exact impact of these segments is unclear, but may be significant [47]. Furthermore, post-transcriptional modifications and sequences themselves may vary with age across different cell types and individuals [48]. The variable expression of RNA-binding proteins may also have an important effect on translation initiation [49–51].

While primary structure close to a putative translation initiation codon remains a strong determinant of translation, the position of a codon within a transcript as well as the secondary structure of the transcript may also have significant impact [52–56]. However, the structure of mRNA(s) associated with repeat expansion disorders is unclear since the exact sequence and varying size of the transcripts are unknown. Moreover, accurate secondary structure requires analysis of a number of nucleotides that far exceeds our incorporated range, and so successful identification of feature patterns may require exceptionally large amounts of training data that are currently unavailable, especially because the number of training samples required to differentiate data increases exponentially as the number of attributes in a model increases [32].

## Enhancing performance

Like other machine learning models, RFC performance is determined by the amount of available balanced training data [57]. Because of this constraint, collecting more examples to train the machine learning models could prove especially useful. In the case of the near-cognate RFC, obtaining sufficient data to account for all near-cognate types could lessen uncertainty in predictions involving these codons. Training the two RFCs discussed here with more data of the codon types that have been used would also be beneficial since feeding a model with more

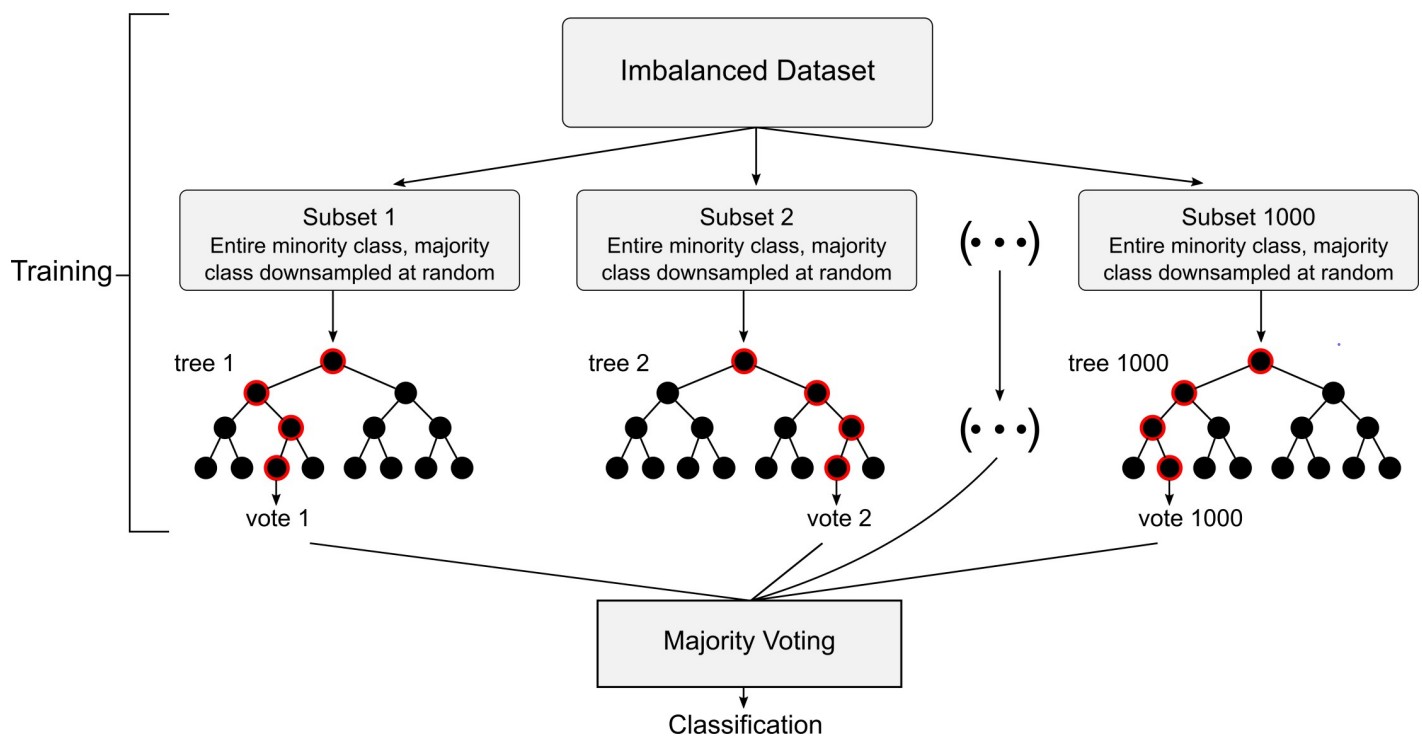

**Fig 15. RFC design.**

data will verify existing trends and introduce variations that the algorithm can recognize and link to a particular classification, thereby improving accuracy.

## Conclusions

Our machine learning algorithms provide insight into translation initiation associated with neurologic repeat diseases. Of note, predictions with high supplementary scores may be associated with significant levels of pathogenic protein production.

## Supporting information

**S1 File. All repeat expansion disorder predictions, with RFC probabilities and KSSs.**
(CSV)

## Acknowledgments

### Accessibility and implementation

The software is publicly accessible as an interactive website at www.tispredictor.com.

Source code, data, as well as models are accessible at https://github.com/Agleason1/TIS-Predictor. A DOI was assigned to the repository using Zenodo: http://doi.org/10.5281/zenodo.5110255.

## Author Contributions

**Conceptualization:** Alec C. Gleason, Raymond P. Roos.

**Data curation:** Alec C. Gleason, Jin Chen, Raymond P. Roos.

**Formal analysis:** Alec C. Gleason.

**Investigation:** Alec C. Gleason, Ghanashyam Ghadge, Yoshifumi Sonobe, Raymond P. Roos.

**Methodology:** Alec C. Gleason.

**Project administration:** Raymond P. Roos.

**Software:** Alec C. Gleason.

**Supervision:** Raymond P. Roos.

**Validation:** Ghanashyam Ghadge, Jin Chen, Yoshifumi Sonobe, Raymond P. Roos.

**Visualization:** Alec C. Gleason, Ghanashyam Ghadge, Jin Chen, Raymond P. Roos.

**Writing – original draft:** Alec C. Gleason.

**Writing – review & editing:** Ghanashyam Ghadge, Jin Chen, Yoshifumi Sonobe, Raymond P. Roos.

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
