## [Decision Letter · Decision Letter 0]

31 Jan 2022

PONE-D-21-24907Machine learning predicts translation initiation sites in neurologic diseases with expanded repeatsPLOS ONE

Dear Dr. Roos,

Thank you for submitting your manuscript to PLOS ONE. After careful consideration, we feel that it has merit but does not fully meet PLOS ONE’s publication criteria as it currently stands. Therefore, we invite you to submit a revised version of the manuscript that addresses the points raised during the review process.

ACADEMIC EDITOR: Based on the comments from the reviewer and my own assessment I recommend major revisions for the article.

We look forward to receiving your revised manuscript.

Kind regards,

Thippa Reddy Gadekallu

Academic Editor

PLOS ONE

https://journals.plos.org/plosone/s/file?id=ba62/PLOSOne_formatting_sample_title_authors_affiliations.pdf”

Reviewers' comments:

Reviewer's Responses to Questions

**Comments to the Author**

1. Is the manuscript technically sound, and do the data support the conclusions?

Reviewer #1: Yes

2. Has the statistical analysis been performed appropriately and rigorously? 

Reviewer #1: Yes

3. Have the authors made all data underlying the findings in their manuscript fully available?

Reviewer #1: Yes

4. Is the manuscript presented in an intelligible fashion and written in standard English?

Reviewer #1: Yes

5. Review Comments to the Author

Reviewer #1: The proposed work presents a machine learning approach for neurologic diseases. It is an exciting research area. However, the following major changes are required.

• I recommend authors to add a separate section to describe the novelties and contributions of the proposed work in detail.

• The comparison of the proposed approach with state of the art is missing. Therefore, a detailed workflow describing the proposed approach is required.

• I recommend authors to use the benchmark dataset and perform similar experiments and discuss the comparison. If a benchmark dataset is not available, authors can create a customized dataset and describe the data collection process in detail.

• The literature review carried out for the proposed work is outdated and needs the referral of some of the latest research works published in the last three years such as https://onlinelibrary.wiley.com/doi/abs/10.1111/exsy.12899, https://www.frontiersin.org/articles/10.3389/fpubh.2021.762303/full, https://www.mdpi.com/1424-8220/20/20/5780.

• I recommend authors to add a layered architecture and detailed work to give more clarity to readers about the proposed system.

• I recommend authors to add limitations in detail(instead of abstract information) of the proposed system and future directions.

• The resolution of all figures is a concern. I recommend authors to redraw most of the images to match the journal standards.

• All tables should be symmetrical and should follow a similar formatting style.

• All the equations should be written using a professional equation editor and should use a similar formatting style and numbering.

6. PLOS authors have the option to publish the peer review history of their article (what does this mean?). If published, this will include your full peer review and any attached files.

Reviewer #1: **Yes: **Sharnil Pandya

---

## [Author Response · Author response to Decision Letter 0]

1 Mar 2022

We thank the academic editor and the reviewer for their comments. Below, we explain our revisions for each point raised. We hope that our updated manuscript now satisfies all criteria and will be suited for publication.

Academic Editor

1. Please ensure that your manuscript meets PLOS ONE's style requirements, including those for file naming. The PLOS ONE style templates can be found at https://journals.plos.org/plosone/s/file?id=wjVg/PLOSOne_formatting_sample_main_body.pdf and https://journals.plos.org/plosone/s/file?id=ba62/PLOSOne_formatting_sample_title_authors_affiliations.pdf”

We have made necessary adjustments to ensure that the manuscript meets PLOS ONE’s style requirements. We followed the requirements for file naming as well.

That is correct. We have not changed our Data Availability statement. Please note that we have also added a file to “Supporting information.”

Reviewer

1. I recommend authors to add a separate section to describe the novelties and contributions of the proposed work in detail.

We have added a subsection under the “Introduction” header called “Novelties and Contributions of the Proposed Work” where we describe the novelties and contributions of the proposed work in detail.

2. The comparison of the proposed approach with state of the art is missing. Therefore, a detailed workflow describing the proposed approach is required.

We have added a subsection in which we compare the performance of our proposed machine learning algorithm to state of the art convolutional neural networks. We further added figures 9 and 15, which outline the workflow of our proposed machine learning algorithms and software.

3. I recommend authors to use the benchmark dataset and perform similar experiments and discuss the comparison. If a benchmark dataset is not available, authors can create a customized dataset and describe the data collection process in detail.

We have added a section in which we compare the performance of the random forest machine learning algorithm to an optimized convolutional neural network using the same training and test datasets. 

4. The literature review carried out for the proposed work is outdated and needs the referral of some of the latest research works published in the last three years such as https://onlinelibrary.wiley.com/doi/abs/10.1111/exsy.12899, https://www.frontiersin.org/articles/10.3389/fpubh.2021.762303/full, https://www.mdpi.com/1424-8220/20/20/5780.

We thank the Reviewer for this suggestion. We have reviewed and cited more research from the last three years, including two of the suggested papers.

5. I recommend authors to add a layered architecture and detailed work to give more clarity to readers about the proposed system.

We have added Fig 9, which depicts the architecture of the software we deploy to visualize predictions. We have also included Fig 15, which details the structure of the random forest classifier incorporated into the system.

6. I recommend authors to add limitations in detail (instead of abstract information) of the proposed system and future directions.

We have added a “Limitations” subsection under “Discussion” which details the limitations of our proposed system. We address future directions in the rest of our discussion.

7. The resolution of all figures is a concern. I recommend authors to redraw most of the images to match the journal standards.

High resolution TIFF images (600 dpi) were originally uploaded with the manuscript; however, they appear blurry on the preprint for reasons beyond our control. The ‘preview pdf’ version of the preprint better displays the images, however, the images have been stretched out.

8. All tables should be symmetrical and should follow a similar formatting style.

All tables are now symmetrical. We have adjusted Table 2 to have a similar formatting style to Table 1.

9. All the equations should be written using a professional equation editor and should use a similar formatting style and numbering.

All equations have been rewritten and numbered using professional equation editor MathType.

---

## [Editor Report · Decision Letter 1]

17 May 2022

Machine learning predicts translation initiation sites in neurologic diseases with nucleotide repeat expansions

PONE-D-21-24907R1

Dear Dr. Roos,

We’re pleased to inform you that your manuscript has been judged scientifically suitable for publication and will be formally accepted for publication once it meets all outstanding technical requirements.

Kind regards,

Thippa Reddy Gadekallu

Academic Editor

PLOS ONE
---

## [Editor Report · Acceptance letter]

20 May 2022

PONE-D-21-24907R1 

Machine learning predicts translation initiation sites in neurologic diseases with nucleotide repeat expansions 

Dear Dr. Roos:

I'm pleased to inform you that your manuscript has been deemed suitable for publication in PLOS ONE. Congratulations! Your manuscript is now with our production department. 

Kind regards, 

on behalf of

Dr. Thippa Reddy Gadekallu 

Academic Editor

PLOS ONE